# Activation of Angiogenic TGF-β1 by Salbutamol Enhances Wound Contraction and Improves Healing in a Streptozotocin-Induced Diabetic Rat Model

**DOI:** 10.3390/cimb47100820

**Published:** 2025-10-03

**Authors:** Promise M. Emeka, Abdulaziz K. Al Mouslem, Hussien Almutawa, Malek Albandri, Hussain Alhmoud, Mohammed Alhelal, Zakaria Alhassan, Abdullah Alhamar

**Affiliations:** Department of Pharmaceutical Sciences, College of Clinical Pharmacy, King Faisal University, Al-Ahsa 31982, Saudi Arabia; aalmoslem@kfu.edu.sa (A.K.A.M.); 221432486@student.kfu.edu.sa (H.A.); 221431058@student.kfu.edu.sa (M.A.); 221428075@student.kfu.edu.sa (H.A.); 221433945@student.kfu.edu.sa (M.A.); zakariaalhassan2000@outlook.com (Z.A.); 219021354@student.kfu.edu.sa (A.A.)

**Keywords:** wound healing, wound contraction, diabetic wound, salbutamol, angiogenesis, TGF-β1

## Abstract

Wound healing is impaired under diabetic conditions due to reduced angiogenesis, thereby increasing the risk of wound-healing complications. Studies have shown that inhibition of α- and β-adrenoceptors delays wound healing. This study investigates the effects of topical salbutamol (TS) on STZ-induced diabetic wound healing in rats. The rats were divided into two initial groups: non-diabetic and diabetic. Diabetes mellitus was induced in the second group with STZ (65 mg/kg). Excision wounds were inflicted on the dorsal thoracic region, 1–1.5 cm away from the vertebral column on either side, following anesthesia on all groups. Group 2 was subdivided into untreated diabetic wounds, low-dose-TS-treated diabetic wounds (6.25 mg/mL), medium-dose-TS-treated diabetic wounds (12.5 mg/mL), and high-dose-TS-treated diabetic wounds (25 mg/mL), and were monitored for 14 days. Percentage wound contraction and the time required for complete wound closure were observed and recorded. In addition, oxidative stress and inflammatory markers such as NO, CRP, MPO, TGF-β1, TNF-α, IL-6, IL-1β, NO, and hexosamine were estimated in wound exudates and tissue over 14 days. TS treatment resulted in 100% wound contraction in all treated wounds within 14 days compared to untreated non-diabetic and diabetic wounds. Increased NO, TGF-β1, and hexosamine activity was observed in TS-treated wounds when compared to untreated diabetic wounds. In addition, TS treatment decreased the activity of IL-1β, TNF-α, IL-6, CRP, and MPO, all of which were elevated in the untreated diabetic wounds. The current study shows that the application of TS significantly improved diabetic wound contraction and aided the healing process. Angiogenic markers, such as TGF-β1 and NO, were prominently increased, supporting the role of sympathetic nerve stimulation in angiogenesis.

## 1. Introduction

Diabetes affects more than 529 million people globally, around 25% of whom experience diabetic wounds [1,2]. The state of hyperglycemia impedes the wound-healing process, resulting in chronic wound phenomena [3]. Importantly, the literature indicates that around 70% of limb amputations occur as a result of unresolved diabetic wounds and are increasing, according to [4]. Dasari et al. [5] reported that diabetes mellitus causes impaired wound healing by interrupting the inflammatory and proliferative phases of wound healing, which leads to a disruptive remodeling phase. Reports indicate that approximately 85 million diabetic patients have developed complications due to delayed wound healing [6].

During the inflammatory phase of wound healing, the body’s hemostatic control begins with debridement and the secretion of cytokines and growth factors [3]. However, diabetic conditions impede angiogenesis, epithelization, and the restoration of damaged tissue integrity, thus impairing wound healing [2]. In a normal wound-healing process, macrophages, which are critical, are typically transformed to produce cytokines like IL-1β, TNF-α, IL-6, and TGF-β. However, in a hyperglycemic state, generated oxidative stress affects macrophage modulation, leading to dysregulation and delayed wound healing [7]. Therefore, non-healing diabetic wounds can result in chronic ulceration with the potential for amputation, particularly in the extremities [6].

One factor known to affect diabetic wound healing is impaired angiogenesis [8]. Documented evidence shows that stimulation of β-adrenergic receptors promotes angiogenesis and tissue growth, aiding in restoration [9]. Further studies have shown that β-adrenergic receptors are abundant in endothelial cells, which play a major role in angiogenesis [10]. Evidence from previous studies indicates that activation of β-adrenergic receptors induces the release of IL-6, stimulating angiogenesis [11]. The authors of [12] reported that isoproterenol significantly improved blood density and flow in an ischemia-induced mouse model. Previous reports also show that overexpression of β-adrenergic receptors restores capillary density and improves angiogenesis [10,13]. Evidence from zebrafish wound healing indicates that the beta 2 adrenergic receptor agonist plays a role in wound inflammation, angiogenesis and, consequently, wound healing [14], and reduces inflammation, therefore acting as a regulator of wound healing/scarring. Given the suggestions that β_2_-adrenergic receptor activation may induce angiogenesis and influence wound healing, we hypothesize that salbutamol, a short-acting β_2_-adrenergic agonist, could have an impact on wound healing, potentially improving angiogenesis, particularly in the context of diabetes-mellitus-induced ulcers. Hence, the present research evaluation seeks to provide valuable insights into the potential therapeutic applications of topical salbutamol for managing diabetic wounds through a study on rats with induced diabetes, exploring its impact on key aspects of the healing process, such as the profiles of cytokines during inflammation, angiogenesis, and wound contraction.

## 2. Materials and Methods

### 2.1. Materials

Streptozotocin (STZ) was purchased from Sigma-Aldrich (St. Louis, MO, USA). We acquired salbutamol sulfate (0.5%) solution (Farcolin-Pharco Pharmaceutical, Alexandria, Egypt) and urethane from Sandoz (Holzkirchen, Germany) and the Accu-Chek^®^ Active Blood Glucose Meter from Roche (Basel, Switzerland).

### 2.2. Methods

#### 2.2.1. Ethical Approval and Animal Procurement

The experimental protocol (No. KFU-REC-2024-DEC-ETHICS2855) was approved by the Institutional Ethics Committee of King Faisal University in compliance with international guidelines for the care and use of laboratory animals. Ethical standards were rigorously followed to ensure animal welfare, and all efforts were made to minimize animal suffering.

A total of 25 rats (aged 4–8 weeks) were procured from the animal housing facility at the College of Clinical Pharmacy, King Faisal University, which were maintained in standardized environmental conditions, including a controlled temperature of 22 ± 2 °C, a relative humidity of 50–60%, and a 12 h light/dark cycle. The animals were housed in individual sterilized cages with ad libitum access to food and water throughout this study.

#### 2.2.2. Grouping and Induction of Diabetes

The animals were divided into two initial groups: one group consisting of 5 rats and the other of 20 rats:

The non-diabetic untreated group (control) comprised 5 rats.

In the diabetic group, diabetes mellitus was induced in 20 rats via a single intraperitoneal injection of STZ at a dose of 65 mg/kg body weight. STZ was freshly dissolved in a cold citrate buffer (pH 4.5) before administration. Blood glucose levels were measured 72 h post-injection using an Accu-Chek^®^ Active Blood Glucose Meter. Rats with fasting blood glucose levels exceeding 200 mg/dL were considered diabetic and included in this study.

### 2.3. Wound Creation and Hemostasis

Standard excision wounds were inflicted on all animals under anesthesia with urethane. The wounds were created on the dorsal thoracic region, positioned 1–1.5 cm lateral to the vertebral column and 5 cm away from the ears. Hemostasis was achieved by gently blotting the wounds with a cotton swab soaked in sterile normal saline. Post-wounding, the animals were housed separately in sterilized, clean cages to prevent infection and cross-contamination and to allow for accurate observation.

### 2.4. Experimental Design and Treatment

The diabetic group was further subdivided into four groups of five rats each, based on the treatment regimen, with two excision wounds per rat, making a total of ten wounds per group.

The untreated diabetic group comprised diabetic rats that did not receive salbutamol treatment.

The low-dose group included 5 diabetic rats treated with topical salbutamol (TS) at a concentration of 6.25 mg/mL.

The medium-dose group comprised 5 diabetic rats treated with TS at a concentration of 12.5 mg/mL.

Finally, the high-dose group contained 5 diabetic rats treated with TS at a concentration of 25 mg/mL, according to a modification of the method of Jemec et al. [15].

The drug was applied topically, directly to the wound area, once daily for 14 consecutive days. The untreated non-diabetic group served as the control for the diabetic groups, receiving no topical application.

### 2.5. Post-Treatment Monitoring

All animals were monitored daily for signs of infection, wound-healing progression, distress, or adverse reactions. Their cages were kept in a sterile condition to minimize the risk of infection and promote healing. Wound healing was assessed periodically using predefined parameters, including wound size reduction, hair growth, and the appearance of granulation tissue. Wound contraction was measured every two post-wounding days using 1mm^2^ graph paper and a transparent sheet according to the method of Yiblte et al. [16]. For accuracy, a thread was used to measure the length and width with the aid of callipers for confirmation. Percent wound contraction was calculated using a formula.

### 2.6. Measurement of the Wound Area and Percentage Contraction

Photos of the wounds were captured on the day they were created (day 1) and daily thereafter until euthanasia on day 14. In addition, wound exudate samples were collected daily before treatment. Sterile cotton swabs, pre-wetted with sterile phosphate-buffered saline (PBS), were used to collect exudate samples from the wounds. Using twisting motions, the swab was rotated 3–5 times using light pressure, covering the entire wound and 2 mm outside of the initial wound edge on each side, and then placed into a microfuge tube with 0.5 mL of sterile PBS and kept on ice. The samples were later stored in a −85 °C freezer until analysis. The circumference of the wound area was measured daily and recorded. The wound closure area was calculated as the ratio of the current wound surface area to the initial wound surface area and expressed in cm^2^ and as a percentage.
$$\mathrm{\% Wound contraction}=\frac{WA_{0}-WAt}{WA_{0}}\times 100\frac{WA_{0}-WAt}{WA_{0}}\times 100$$

### 2.7. Wound Standardization

Wound standardization was carried out with each wound acting as its own control. Therefore, initial size for a particular wound, was monitored for that particular wound in the group. The contraction was calculated based on its initial wound size as respective sizes of the initial wounds, no matter the group, serves as it control for that wound. Hence, that wound is monitored till the end based on the initial wound size only. By so doing the extent of wound contraction from that particular wound is calculated based on its initial wound size, and accurately measure the actual size of contraction from the initial wound excision.

### 2.8. Biochemical Analysis

Wound tissue samples were collected after euthanizing the mice. NO, CRP and myeloperoxidase (MPO, A003-1-2) activities were assessed in wound exudate and tissue homogenates. In all biochemical tests, total protein was measured and utilized as an internal control. Nitrites and nitrates are formed as reactive nitrogen end-products during NO formation and were measured using Griess reagent. ELISA kits were used to quantify their levels in the homogenates.

### 2.9. Hexosamine

Homogenized tissue (0.05 mL) was diluted to 0.1 mL with 0.1M phosphate-buffer solution (PBS). To this, 0.1 mL of acetyl acetone reagent was added, heated in a boiling water bath for 20 min, and then cooled under tap water. To this, 1.5 mL of 95% alcohol was added, followed by an addition of 0.5 mL of Ehrlich’s reagent. The reaction was allowed to stand for 30 min until completion. Color intensity was measured at 530 nm against the blank. The hexosamine content of the samples was determined from the standard curve prepared with D (+) glucosamine hydrochloride (HiMedia Laboratories Pvt. Ltd., Mumbai, India), from 5 to 50 μg/0.5 mL, using a 100 μg/mL working solution.

### 2.10. Cytokine Estimation

The cell-free supernatant of wound exudate and dermal tissues was used for the estimation of cytokines such as IL-1B, IL-6, TNF-a and TGF-β1. Their concentrations in the different homogenate samples were estimated using enzyme-linked immunosorbent assay (ELISA) kits, and the values were expressed as pg/mL (Genway, San Diego, CA, USA).

### 2.11. Molecular Docking Analysis

The AutoDock (v4.2) and AutoDock Tools (ADT, v1.5.4) programs were used to perform the docking analysis [17]. Salbutamol’s (3D) chemical structure (CID_2083) was retrieved from the PubChem database. The three-dimensional structures of the target protein TGFB1 (PDB ID: 9fdy) for mice were obtained from http://www.pdb.org (https://www.rcsb.org/structure/9FDY, assessed on 24 May 2025). Target protein complexes were docked with a flexible-bodied ligand and a rigid-bodied molecule. We searched for the whole-receptor amino acid employed for blind docking using the Lamarckian Genetic Algorithm; 150-person populations with a 0.02 mutation rate evolved over the course of five generations. The various complexes were sorted according to the anticipated binding energy in order to assess the outcomes. Then, a cluster analysis was carried out using root-mean-square deviation values in relation to the initial geometry. The solution with the lowest energy conformation in the most populous cluster was determined the most reliable. The trial version of Discovery Studio 2021 Client was used to visualize and analyze docked ligand–receptor interactions.

### 2.12. Statistical Analysis

Data are presented as mean ± SD, and statistical analysis was performed using the GraphPad Prism software (version 10.4.1) to determine statistical significance, employing two-way and one-way ANOVA. The Tukey multiple-comparison test was used to compare between and among the studied groups. Proportions were compared using the MedCalc Software Ltd. Comparison of Proportions Calculator (version 23.2.1; accessed on 1 April 2025 at https://www.medcalc.org/calc/comparison_of_proportions.php). A *p*-value < 0.05 was considered statistically significant.

## 3. Results

### 3.1. Wound Contraction

Table 1 shows the topical application of salbutamol solution (TS) to healing excision wounds in rats with induced diabetes, describing the progression of wound contraction induced by TS. The results demonstrated that its administration significantly facilitated wound contraction at all dosage levels from day 3 to day 14 compared to the control. As depicted in Table 1, wound contraction in the group treated with TS occurred at a significantly increased rate compared to both the untreated diabetic and control groups.

On day 3, the untreated diabetic group showed significantly smaller wound contraction (8.3%) compared to the non-diabetic untreated group, which exhibited 17% wound closure (*p* < 0.05). Treatment with TS produced significantly greater wound contraction on day 3 compared to the untreated diabetic wound with 26.7, 20.9 and 23.5% contraction (Table 1). On day 7, wound contraction in the treatment groups was better than in the non-diabetic control, which had 41% contraction, whereas wound contraction with TS treatment showed a dose-dependent response, (53, 54, and 57.7%, respectively), with the higher doses resulting in markedly greater contraction than in untreated diabetic wounds (34%) However, only the higher dose showed a significant difference (*p* < 0.005) when compared to both the non-diabetic and untreated diabetic groups.

On day 14, all diabetic wounds treated with TS achieved 100% wound closure, whereas untreated diabetic wounds had 83% wound contraction, and the control group exhibited 91% contraction. These findings indicate that treatment with TS significantly promoted wound closure compared to both untreated diabetic and non-diabetic control wounds. The dynamics of wound healing in the treatment groups are obviously different and faster due to application of TS.

Furthermore, Figure 1 illustrates the pictorial progression of wound contraction and eventual closure. The treated groups achieved complete healing by day 14, while both the untreated diabetic and control groups required 20 days to achieve complete healing.

### 3.2. Oxidative Markers Involved in Wound Healing

Figure 2A,B represents the levels of oxidative markers involved in wound healing, including nitric oxide (NO), C-reactive protein (CRP) from wound exudate swabs, and myeloperoxidase (MPO) from wound tissue samples, on days 1, 3, 7, and 14. The levels of NO, CRP, and MPO were compared across different experimental groups on day 14.

#### 3.2.1. Wound Exudate

The results showed increased production of NO on day 3 for all treatment groups, which was significantly higher than in the untreated diabetic group (*p* < 0.001) in the wound exudate analyzed. However, no significant difference in NO levels was observed between the non-diabetic control and treated diabetic groups on day 14. In addition, as wound contraction increased, NO levels declined, indicating a correlation between wound closure and NO levels (Figure 2A).

As shown in Figure 2B, CRP levels in wound exudate were significantly higher (*p* < 0.001) in untreated diabetic wounds compared to both the non-diabetic control and all treated diabetic groups. These results suggest lower CRP activity with treatment using TS. Moreover, the high-dose-treated group exhibited the lowest levels of CRP on day 14 compared to the other treatment groups and control.

#### 3.2.2. Wound Tissue Oxidant Markers

As shown in Figure 3A,C, the levels of NO measured in the tissue of diabetic excision wounds correlated with levels found in wound exudate samples. Untreated diabetic wounds generated minimal NO, with low levels observed on days 1 and 3. However, NO levels were significantly higher (*p* < 0.001) in the medium- and high-dose-treated wounds compared to untreated diabetic wounds. Furthermore, on day 14, untreated diabetic wounds showed an increase in NO levels, but this was not statistically significant compared to the low-dose-treated wounds. A dose-dependent effect of TS treatment on diabetic wounds was also observed.

On day 14, CRP levels, determined from excised diabetic wound tissue, were elevated in untreated diabetic wounds, which correlated with the results obtained from the wound exudate samples. This finding was highly significant (*p* < 0.0001). The high-dose-treated wounds exhibited the lowest tissue CRP levels, as shown in Figure 3B, and these levels were even lower than those observed in the control group. Additionally, there was no significant difference in CRP levels between the low-dose and medium-dose treatments.

Figure 3C displays the wound tissue MPO activity for different treatment groups on day 14. The data show that untreated diabetic wounds had the highest MPO activity, with a significant difference (*p* < 0.005) compared to the medium- and high-dose-TS-treated diabetic wounds. Although no significant difference in MPO activity was observed between the low-dose-TS-treated diabetic wounds and untreated diabetic wounds, the MPO activity in control wounds was significantly lower (*p* < 0.0001). In this evaluation, treated diabetic wounds exhibited reduced MPO activity in a dose-dependent fashion, with no significant difference when compared to the non-diabetic untreated control wounds.

### 3.3. Levels of Angiogenic Marker Transforming Growth Factor Beta 1 (TGF-β1)

The results revealed that untreated diabetic wounds produced consistently low levels of TGF-β1, whereas the control and all treated diabetic wounds showed increased levels from day 3. However, TGF-β1 levels began to decrease by days 7 and 14, correlating with the progression of wound healing past the angiogenesis phase (Figure 4). Although TS-treated wounds exhibited a dose-dependent increase, the high-dose TS treatment produced the highest increase in tissue TGF-β1 levels from day 1 to day 14. The non-diabetic untreated control group had levels similar to those observed in the low-dose-treated wounds.

### 3.4. Proinflammatory Cytokine Concentrations

Figure 5A–C describe the levels of IL-1β, TNF-α, and IL-6 in wound tissue homogenates from untreated non-diabetic control, untreated diabetic, and TS-treated diabetic wounds. IL-1β is known to be induced during the inflammatory phase of wound healing and showed significantly higher levels (*p* < 0.001) in untreated diabetic wounds compared to all TS-treated wounds. The highest dose of TS resulted in very low IL-1β levels, as shown in Figure 5A, with a dose-dependent decrease in IL-1β levels, significantly lower compared to untreated diabetic wounds (*p* < 0.0001). In addition, the non-diabetic untreated control wounds had significantly higher IL-1β levels compared to medium- and high-dose-TS-treated wounds on days 1 and 3 (*p* < 0.002). Among the TS-treated wounds, the low-dose TS treatment produced the highest IL-1β levels (Figure 5A).

Regarding tissue levels of TNF-α, as presented in Figure 5B, the non-diabetic untreated control wounds produced the lowest levels observed in this investigation. Furthermore, untreated diabetic wounds exhibited significantly higher levels of TNF-α from days 1 to 7 (*p* < 0.001). On day 1, high-dose-TS-treated wounds had relatively high TNF-α levels, but these gradually decreased from day 7 to day 14. This resulted in no significant difference between the control and high-dose TS treatment.

On the other hand, inflammatory cytokines evaluated in wound tissue samples on day 14 indicated that the non-diabetic untreated control had the lowest IL-6 levels. This was found to be significant (*p* < 0.001) compared to the untreated diabetic wounds, which had a high increase. Therefore, untreated diabetic wounds characteristically induced a high level of IL-6. Among the TS-treated wounds, IL-6 levels decreased in a dose-dependent manner, indicating that treatment with TS reduced the activation of IL-6, an inflammatory inducer during tissue injury (Figure 5C).

TNF-α(measured from day 1 to 14) and IL-6 (measured only on day 14) were analyzed from untreated non-diabetic control, untreated diabetic and TS-treated wounds. *** represents significant difference at *p* < 0.0001 for IL1-beta and TNF-alpha, whereas for IL-6, data are presented as mean ± SD, *n* = 4. * represents significant difference at *p* < 0.05, ** denotes significant difference at *p* < 0.005, and *** represents *p* < 0.0005, whereas **** represents significant difference at *p* < 0.0001; ns = not significant; TS = topical salbutamol.

### 3.5. Hexosamine Activity in Diabetic Excision Wounds

The results of hexosamine analysis in diabetic wound tissue homogenates after 14 days of observation revealed significantly lower activity (*p* < 0.001) in the untreated diabetic wounds compared to the untreated non-diabetic controls. In addition, lower hexosamine activities were recorded for all TS-treated wounds. However, no significant differences were seen between TS-treated wounds and the untreated non-diabetic control. Notably, high-dose-TS-treated wounds exhibited a highly significant difference (*p* < 0.002) compared to untreated diabetic wounds (Figure 6).

### 3.6. Molecular Docking of Salbutamol and TGF-β1

To further evaluate the potential angiogenetic mechanism of salbutamol, its interactions with TGF-β1 and its receptor were assessed by performing molecular docking. The binding of salbutamol to the TGF-β1 receptor was found to be stable, as shown in Figure 7A,B, with the acceptor area in green and hydrogen bonding in pink. Therefore, the binding of salbutamol appears to be attributable to its hydrogen bond interaction between Ala 75 (3.25/3.10) and Gly 46 (2.97), as shown in Figure 7C and Table 2. The stability of salbutamol binding to TGF-β1 also exhibited an estimated binding energy of −5.96, with an intermolecular energy of −6.95.

## 4. Discussion

Diabetic wound healing is often complicated by induced oxidative stress, leading to reduced antioxidant activity due to hyperglycemia [18]. During hyperglycemia, wounds are rapidly assaulted by inflammatory mediators, which delay the healing process. This could potentially lead to chronic wound development [19]. Drugs used in diabetic wound healing are scarce, with only Becaplermin gel available. Becaplermin gel is the only approved growth factor treatment for diabetic foot ulcers and has been shown to promote wound healing [20]. However, safety concerns have been issued by the FDA in the form of black box warning about a potential malignancy risk, amidst other contraindications. Furthermore, a systemic review indicates that the safety data were poorly reported and adverse events may have been underestimated [21]. Therefore, the present study only used an untreated control as the benchmark.

Previous reports showed that salbutamol attenuated inflammatory mediators and was found to reduce IL-1β levels [22] and exhibit anti-inflammatory effects in experimental carrageenan-induced inflammation [23].

β-adrenoceptors are well-known G-protein-coupled receptors reported to be involved in neoangiogenesis [10]. Hence, the stimulation of *β*2-adrenoceptors is likely linked to the expression of endothelial progenitor cells associated with angiogenesis [24,25]. A previous study also showed that increased expression of *β*2-adrenoceptors resulted in enhanced angiogenic activity in endothelial progenitor cells [26]. According to Xanthopoulos et al. [9], antagonism of *β*-adrenoceptors inhibited cellular angiogenesis, further supporting the role of *β*2-adrenoceptor agonists.

Proper wound healing requires strong blood circulation, which is facilitated by angiogenesis [22]. Pan et al. [27] reported that sympathetic nerves play a significant role in soft tissue angiogenesis. Their study suggested that stimulation of the sympathetic nervous system promotes angiogenesis, while its blockade will lead to delayed wound healing in rats [28]. Another study using terbutaline, a *β*2-adrenergic agonist, improved angiogenesis [25].

A previous study [14] showed that TS improved wound scarring and reduced pigmentation in a zebrafish wound model. The present study evaluated the potential effects of TS on excision wound healing in an STZ-induced diabetic rat model.

Wound contraction was observed from day 3 of treatment with TS, with a dose-dependent increase in contraction observed by day 7. Wound closure was completed for all TS-treated wounds within 14 days of observation, aligning with the suggestion that TS promotes angiogenesis, which is crucial for wound healing [27]. Hence, angiogenesis facilitates the development of new blood vessels delivering oxygen and nutrients with a consequent removal of debris. It is documented that diabetic conditions often impair angiogenesis due to sympathetic nerve damage and subsequent reduced blood flow to certain areas of the body [29].

We examined the roles of oxidant and proinflammatory mediators in the context of treating excision wounds in an induced diabetes model. NO, which was analyzed from wound exudates, was found to be lower in the untreated diabetic wounds compared to the higher levels observed in TS-treated wounds. Previous studies indicate that NO generation is important for wound healing and generally increases during the inflammatory phase of healing [30]. Our findings are consistent with this report, confirming the results of previous studies. Furthermore, NO levels in wound exudates correlated with those in wound tissue homogenates, reflecting a similar trend across all study groups.

Increased CRP levels were observed in wound exudates from the untreated diabetic wounds. CRP is known to be elevated in diabetic conditions [31]. Yu et al. [32] also documented a similar increase in CRP in diabetic rats, identifying it as an inflammatory marker. Prolonged elevation in CRP is known to impair wound healing [32]. Our analysis of CRP from wound tissue homogenates on day 14 showed similar trends, with high levels found in untreated diabetic wounds. Previous studies suggest that wound exudates can serve as valuable diagnostic tools for monitoring wound healing and therapy [33].

Evidence from previous studies revealed a dynamic interaction between angiogenic cytokines [22]. This associative interplay is observed among TGF-β1, vascular endothelial growth factor, and the extracellular matrix in modulating angiogenesis [22].

Deng et al. [18] suggested that increased levels of TGF-β1 can enhance diabetic wound healing. This study demonstrated increased levels of TGF-β1 in TS-treated wounds, along with a reduction in its levels in untreated diabetic wounds. Evidence has shown that TGF-β1 possesses angiogenic activity [22,34], thus improving wound healing, and that collagen promotes the activity of TGF-β1 in wound healing, thereby decreasing wound areas faster and thus accelerating the wound-healing process [35]. Furthermore, it is known that an increase in tissue NO content can elevate TGF-β1 levels and promote angiogenesis. Therefore, TS may enhance TGF-β1 levels by increasing tissue NO content.

Other proinflammatory markers and cytokines, such as IL-1β, TNF-α, and IL-6, were evaluated in this study and found to be elevated in the untreated diabetic wound. These findings are consistent with previous reports showing that elevated levels of these proinflammatory markers are associated with delayed wound healing and impaired angiogenesis [36,37]. Therefore, sustained activation of the IL-1β pathway in untreated diabetic wounds may contribute to impairments in the healing process, as the release of IL-1β regulates inflammatory cytokines. This assertion is supported by studies showing that inhibiting IL-1β signaling in diabetic wounds can improve wound healing [38]. However, TS treatment significantly reduced the levels of IL-1β, TNF-α, and IL-6, in agreement with the findings of [11].

Hexosamine activity was decreased in untreated diabetic wounds after 14 days, as indicated by wound tissue homogenates. However, TS-treated wounds exhibited increased hexosamine activity, suggesting that TS may aid wound closure. These findings are consistent with those of [11], as studies have shown that hexosamine activity is reduced in diabetic conditions and increased in non-diabetic situations, as observed in this study [39]. Another previous study has shown that hexosamine synthesis stimulates wound repair and healing [40].

In evaluating the factors responsible for TS-induced wound healing, MPO activity, being activated as an early sign of inflammation [33], was also assessed. As expected, it was significantly elevated in untreated diabetic wounds but reduced following TS treatment. Notably, high-dose TS treatment led to similar MPO levels in comparison to the non-diabetic untreated wounds, indicating that TS treatment may normalize MPO activity. Molecular docking analysis revealed that salbutamol binds to the TGF-β1 receptor, indicating potential to act via this pathway. This could be the reason why TGF-β1 was elevated during treatment with TS, in line with the angiogenetic activity of sympathetic stimulation.

These findings support previously documented evidence that TS plays a role in wound healing, particularly in diabetic wounds.

## 5. Study Limitations

The main limitation of this study is its lack of analysis of the expression levels of various proteins involved in angiogenesis, such as via Western blotting or real-time PCR. In addition, we did not assay wound bacteria or perform microscopic wound analysis due to limited funding.

## 6. Conclusions

The present study evaluated the use of TS on STZ-induced diabetic wounds and found that its application enhanced the levels of TGF-β1 and NO, two complementary mediators that play an active role in angiogenesis. We observed an increase in hexosamine activity, suggesting early wound contraction in the diabetic wounds evaluated. Moreover, the activities of inflammatory mediators responsible for delayed wound healing, such as IL-1β, TNF-α, IL-6, CRP, and MPO, which are usually increased under diabetic conditions, were all reduced; this could thus have contributed to the improved wound healing observed in this study and supports the potential use of TS as a therapeutic tool. Molecular docking confirmed the interaction between salbutamol and TGF-β1, indicating a possible activation and confirming its angiogenic potential. Although TS’s mechanism of action appears to be multifactorial due to its interaction with multiple inflammatory pathways, further studies are needed to fully elucidate this precisely in the context of wound contraction and healing.

## Figures and Tables

**Figure 1 cimb-47-00820-f001:**
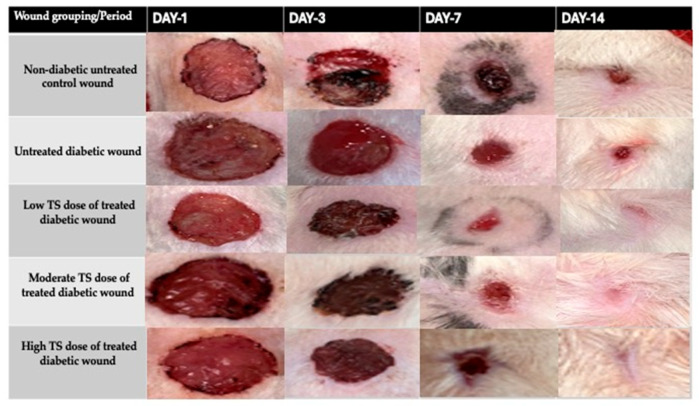
Photographic illustrations of the wound-healing process in untreated non-diabetic control, untreated diabetic, and treated diabetic wounds on different days of observation following TS treatment.

**Figure 2 cimb-47-00820-f002:**
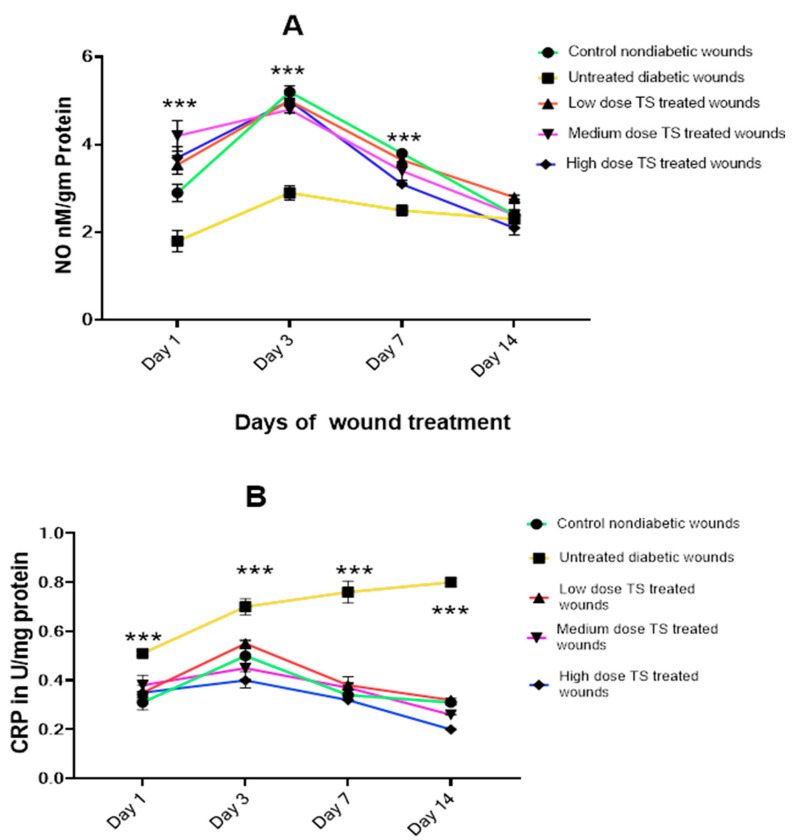
(**A**,**B**) represent the levels of oxidative markers involved in wound healing: NO and CRP from wound exudate samples, measured on days 1, 3, 7, and 14. Data are presented as mean ± SD, *n* = 4. *** represents significant difference, *p* < 0.005. TS = topical salbutamol.

**Figure 3 cimb-47-00820-f003:**
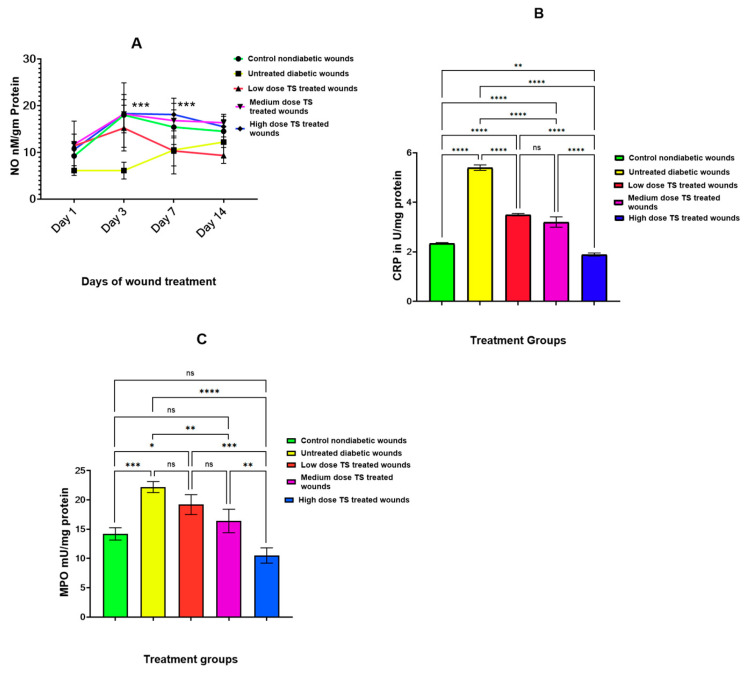
(**A**–**C**) represent the levels of oxidative markers involved in wound healing: NO (measured on days 1, 3, 7, and 14), CRP and MPO from wound tissue samples measured on day 14. Data are presented as mean ± SD, *n* = 4. * represents significant difference *p* < 0.05, ** denotes significant difference at *p* < 0.005, and *** represents *p* < 0.0005, whereas **** represents significant difference at *p* < 0.0001. TS = topical salbutamol; ns = not significant.

**Figure 4 cimb-47-00820-f004:**
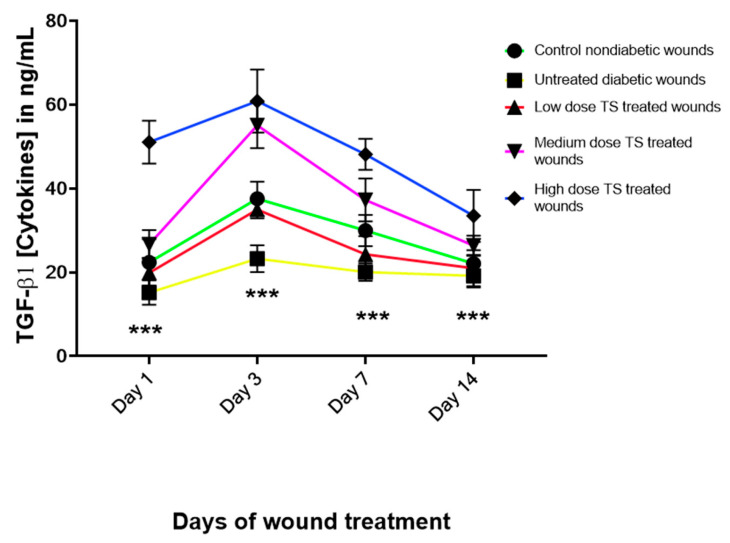
Results shown in this figure represent the angiogenetic maker, TGF-β1, analyzed from tissues sampled on days 1, 3, 7, and 14. Data are presented mean ± SD, *n* = 4. *** represents significant difference at *p* < 0.0001. TS = topical salbutamol.

**Figure 5 cimb-47-00820-f005:**
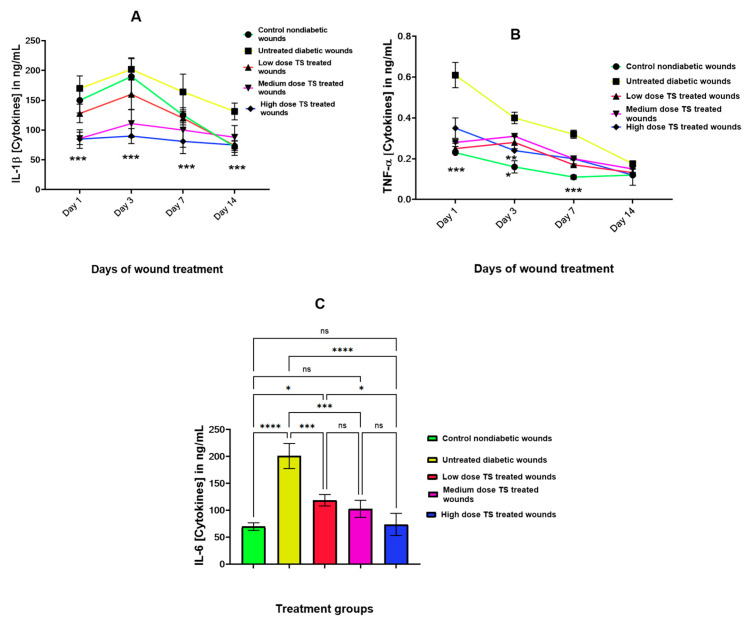
(**A**–**C**) show levels of proinflammatory cytokines in wound tissue homogenates. Concentrations of IL-1β, TNF-α (measured from day 1 to 14) and IL-6 (measured only on day 14) were analyzed from untreated non-diabetic control, untreated diabetic and TS-treated wounds. *** represents significant difference at *p* < 0.0001 for IL1-beta and TNF-alpha, whereas for IL-6, data are presented as mean ± SD, *n* = 4. * represents significant difference at *p* < 0.05, ** denotes significant difference at *p* < 0.005, and *** represents *p* < 0.0005, whereas **** represents significant difference at *p* < 0.0001; ns = not significant; TS = topical salbutamol.

**Figure 6 cimb-47-00820-f006:**
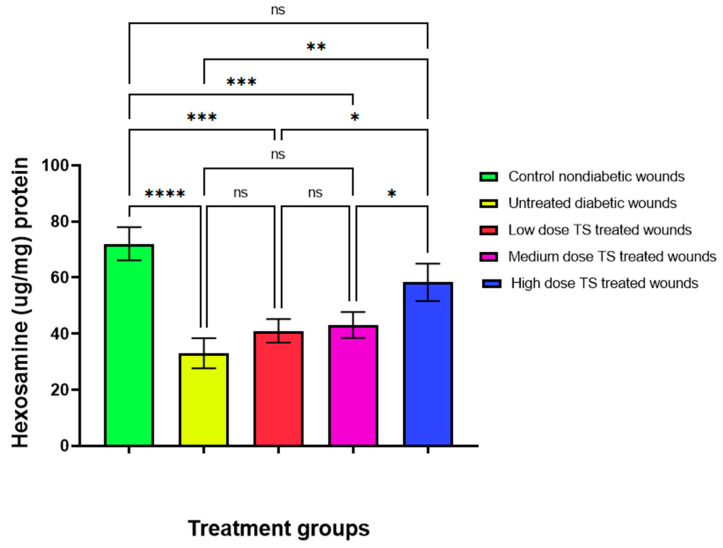
Levels of hexosamine in wound tissue homogenates from the untreated non-diabetic control, untreated diabetic, and TS-treated diabetic wounds after 14 days of observation. Data are presented as mean ± SD, *n* = 4. * represents significant difference *p* < 0.05, ** denotes significant difference at *p* < 0.005, and *** represents *p* < 0.0005, whereas **** represents significant difference at *p* < 0.0001. TS = topical salbutamol; ns = not significant.

**Figure 7 cimb-47-00820-f007:**
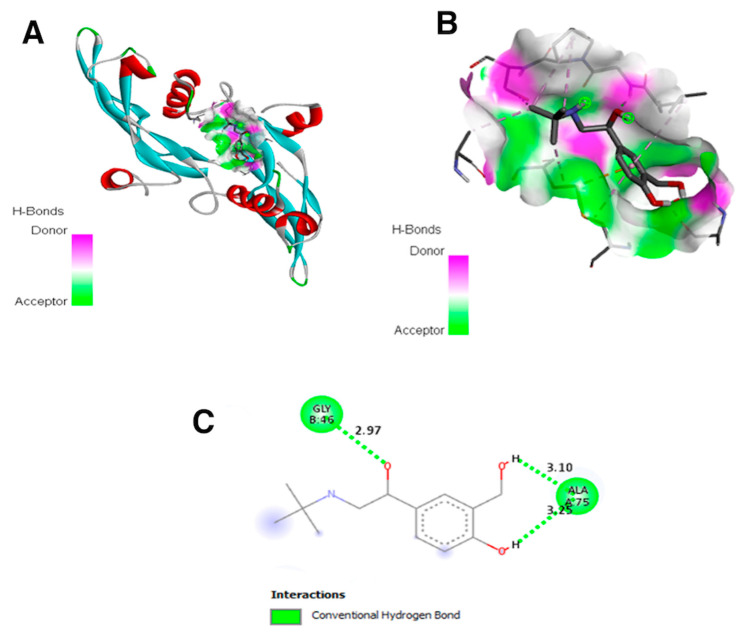
Interaction of salbutamol with the active site of TGF-β1. Green indicates the involvement of the hydrogen bond in the salbutamol–TGF-β1 interaction. (**A**) represents 2D interaction; (**B**) represents 3D structure interaction. (**C**) shows the binding of salbutamol with hydrogen bond interaction between Ala 75 (3.25/3.10) and Gly 46 (2.97).

**Table 1 cimb-47-00820-t001:** Percentage (%) of wound contraction in non-diabetic control and various treatment groups from day 1 to day 14.

Treatment Groups	Day 1	Day 3	Day 7	Day 14
Non-diabetic Control	A = 1.85 ± 0.58 cm^2^ WC = 0%	A = 1.53 ± 0.55 cm^2^ WC = 17% *	A = 1.081 ± 0.34 cm^2^ WC = 41%	A = 0.175 ± 0.10 cm^2^ WC = 91%
Untreated diabetic	A = 1.68 ± 0.36 cm^2^ WC = 0%	A = 1.54 ± 0.38 cm^2^ WC = 8.3%	A= 1.111 ± 0.19 cm^2^ WC = 34%	A = 0.291 ± 0.2 cm^2^ WC = 83%
Low-dose TS	A = 1.92 ± 0.33 cm^2^ WC = 0%	A = 1.41 ± 0.09 cm^2^ WC = 26.7% ***	A = 0.9 ± 0.11 cm^2^ WC = 53.18% **	WC = 100% ***
Medium-dose TS	A = 1.95 ± 0.059 cm^2^ WC = 0%	A = 1.54 ± 0.2 cm^2^ WC = 20.9% *	A = 0.895 ± 0.11 cm^2^ WC = 54.1% **	WC = 100% ***
High-dose TS	A = 2.6 ± 0.39 cm^2^ WC = 0%	A = 1.99 ± 0.01 cm^2^ SWC = 23.5% **	A = 1.1 ± 0.07 cm^2^ WC = 57.7% ***	WC = 100% ***

A = Total wound area in cm^2^; WC = wound contraction; TS = topical salbutamol. * = *p* < 0.05; ** = *p* < 0.005; *** = *p* < 0.0005.

**Table 2 cimb-47-00820-t002:** Interactions of salbutamol and amino acid residues of TGFB1 (PDB ID: 9fdy).

Ligand	PubChem ID	Binding Energy	Ligand Efficiency	Intermolecular Energy	Docked Amino Acid Residue (Bond Length)
Salbutamol	2083	−5.96	−0.35	−6.95	Chain A: ALA′75′(3.25 Å) Chain A: ALA′75′(3.10 Å) Chain B: GLY′46′(2.97 Å)

## Data Availability

The raw data supporting the conclusions of this article will be made available by the authors on request.

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
