# Peer review of "Activation of Angiogenic TGF-β1 by Salbutamol Enhances Wound Contraction and Improves Healing in a Streptozotocin-Induced Diabetic Rat Model"

_cimb, 2025, doi:10.3390/cimb47100820_

Round 1

Reviewer 1 Report

Comments and Suggestions for Authors

Dear Authors,

Your study undoubtedly addresses a highly relevant topic. However, I have several suggestions that I believe could significantly improve your manuscript.

Introduction (Minor):

  1. The section lacks a reference stating that salbutamol is a short-acting β2-adrenergic agonist (lines 61–63). Additionally (at the authors’ discretion), it might be helpful to elaborate further on why salbutamol was chosen as the therapeutic agent.

Materials and Methods (Major Revision Required):

The current description of the experimental procedure appears insufficient for reproducibility. Please address the following:

  1. Were all animals susceptible to STZ? Were there any unexpected fatalities? Was blood glucose monitored throughout the experiment? On what basis was the specific treatment (TS) dose selected?

  2. Please clarify how the initial wound size was standardized across animals.

  3. What was the rationale behind selecting the three treatment concentrations of TS?

  4. How exactly was wound measurement performed? This needs a highly detailed description in the relevant section.

  5. How was wound exudate collected? Please specify the procedure.

  6. "Dermal tissue samples were collected after euthanizing the mice." (lines 128) – This may be a typo; please verify. Also, specify the kits used in this section.

  7. Additionally, please state the formula used to calculate wound closure.

Results (Major):

  1. Please standardize the abbreviation for salbutamol (either Sal or TS).

  2. While a matter of choice, presenting wound closure as a graph might improve clarity. Also, include a scale (e.g., a ruler) in the wound photographs to assess size.

    Notably, in the provided images (Day 14), wounds remain visually apparent across all treatment groups—while moist scabs are absent, the boundaries are still discernible. I question the reported 100% healing and urge a thorough explanation of:

    • The measurement method (including software used).

    • How wound size was normalized in photographs.

    (That said, visual estimation suggests ~98% closure, so my comment does not detract from the promising results—it’s chiefly about methodology transparency.)

  3. The absence of a positive control group (e.g., Regranex/growth factors/etc) should be discussed in-depth, contextualizing findings within existing literature data.

  4. Consider converting Figure 2 to color for better readability.

  5. The phrasing in lines 212–213 seems redundant; please review.

  6. The fragmentation of Figure 2 is unconventional. I recommend continuous numbering (e.g., relabeling panels 2C–E as Figure 3A–C). Color adjustments (as above) would also help.

Discussion (Major):

  1. Emphasize the novel contributions of your work.

  2. Explain how salbutamol (a bronchodilator) might influence angiogenesis. Could this be a nonspecific positive effect? Address this by:

    • Acknowledging the lack of nonspecific controls (e.g., exogenous protein).

    • Expanding the discussion with relevant literature data.

  3. Clearly state the limitations of your study.

Reviewer 2 Report

Comments and Suggestions for Authors

Thank you for the chance to have review interesting paper. Author argued salbutamol effect on  wound contraction and healing. I have several concerns.

1. Salbutamol were classified as b2-adrenergic receptor agonist. Please make it clear how b2-adrenergic receptor activation affect on wound healing, or clarify novel molecular mechanism of the salbutamol in wound healing process.

2. It was interesting to investigate docking simulation of salbutamol to TGFb1, not TGFb receptor. Please add details why author tired to in-silico binding simulation on Ligand, rather than receptor.

3. Among 30 references, 9 articles were published within 5 years. Please update references.

4. Figure images were unclear. Please improve resolution of the figures.

Reviewer 3 Report

Comments and Suggestions for Authors

Comments

The manuscript entitled “Activation of Angiogenic TGF-β1 by Salbutamol Enhances Wound Contraction and Improves Healing in Streptozotocin-Induced Diabetic Rat Model” addresses an important clinical issue: impaired wound healing in diabetes. The work is interesting and potentially relevant, but several issues need to be addressed before the manuscript can be considered for publication.

Major concerns

  1. The Control and untreated diabetic groups should receive the vehicle treatment to ensure proper comparison with salbutamol-treated groups.
  2. The molecular docking method must be described in the Materials and Methods section in sufficient detail for reproducibility.
  3. Table 1 is challenging to interpret. Please revise for clarity or present the data in graphical form for better understanding.
  4. Figure 1: The wounds appear irregular. A biopsy punch is recommended for consistent wound size and reproducibility. Please clarify how wounds were created.
  5. Figure 2 contains multiple overlapping panels. Please either combine the graphs on one page or separate panels C–E into a new figure. Revise the figure legends accordingly.
  6. Figure 2C: It is unclear whether rats were sacrificed at each time point to measure NO from wound tissue. If not, creating new wounds would confound healing outcomes. If yes, please clarify the total number of animals used.
  7. The study lacks comparison with established wound-healing therapies. For example, recombinant human PDGF (becaplermin gel) is clinically approved for diabetic wounds, and topical insulin therapy has also been reported to promote healing. Including such comparisons, or at least discussing them, would provide greater translational context.
  8. The Discussion essentially reiterates the results. Please expand by comparing your findings with previous studies on β2-adrenergic agonists in wound healing.
  9. Please address potential safety concerns of salbutamol (systemic absorption, cardiovascular effects) when used in diabetic wounds, as this would be highly relevant for translational application.

Minor concerns:

  1. Line 48-49: This sentence should be moved to the previous paragraph, following the epidemiology of diabetes with impaired wound healing.
  2. Figure legend: Please indicate the sample numbers (n) for each dataset in the legends.
  3. Lines 212-213 appear to duplicate information from the Figure 2 legend. Please revise.

Round 2

Reviewer 1 Report

Comments and Suggestions for Authors

Dear authors,

I still have some questions regarding the wound healing processes described in your study. Could you please clarify the following points?

  1. A group of non-diabetic animals cannot be considered a positive control. By definition, a positive control should involve an agent with established efficacy, such as Regranex (c). I understand that including such a group may pose some challenges, but I strongly recommend addressing this limitation thoroughly in the discussion section of your results.
  2. Using a string and ruler to measure wounds is not a standardized method. The standard approach involves calculating wound area, which can be done in any software capable of scale-based measurement (e.g., ImageJ). To implement this, you set the scale using the ruler photographed alongside the wound, then measure the area relative to that scale.

    From my preliminary assessment of your images, the wounds in non-diabetic animals appear larger than those in diabetic animals. This raises critical questions: Does wound healing proceed more rapidly in the streptozotocin-induced model, or are the healing dynamics actually comparable to those in non-diabetic controls? This section of the results needs thorough revision to resolve these inconsistencies.

  3. I would also like to highlight the wounds treated with varying TS doses. While these wounds do show noticeable improvement—lacking moist crusts and appearing less severe—it would be incorrect to claim complete healing, as the wounds remain clearly visible.

  4. How were the wound sizes standardized in this study? The demonstration photographs show considerable variation in initial wound sizes. Including a ruler or scale bar in these images would have been helpful for accurate comparison. Visually, the wounds in the STZ group seem almost twice as large as those in the untreated control animals. Moreover, by day 14, the wounds in the STZ group appear smaller than those in the untreated controls. There is also noticeable variability in wound size across different concentrations of TS.
    Given these observations, this part of the study requires significant revision to ensure consistency and reliability in the results.

Reviewer 3 Report

Comments and Suggestions for Authors

The authors have addressed the concerns, and the manuscript is suitable for publication.

Author Response

Authors would like thank you for your valuable imput to the the making of this manuscript.

We have addressed some issues raised by one of the reviewers to improve the results further. All  figures have been improved converted to colour.

Round 3

Reviewer 1 Report

Comments and Suggestions for Authors

Dear authors,

Thank you for addressing all my comments. At this point, I have no further suggestions.